# Quantifying and Reducing Ion Migration in Metal Halide Perovskites through Control of Mobile Ions

**DOI:** 10.3390/molecules28135026

**Published:** 2023-06-27

**Authors:** Saivineeth Penukula, Rodrigo Estrada Torrejon, Nicholas Rolston

**Affiliations:** School of Electrical, Computer and Energy Engineering, Arizona State University, Tempe, AZ 85281, USA; spenukul@asu.edu (S.P.); raestra6@asu.edu (R.E.T.)

**Keywords:** perovskite solar cells, mobile ion concentration, ionic mobility, stability, defects, vacancies, degradation, activation energy, impedance spectroscopy, transient current

## Abstract

The presence of intrinsic ion migration in metal halide perovskites (MHPs) is one of the main reasons that perovskite solar cells (PSCs) are not stable under operation. In this work, we quantify the ion migration of PSCs and MHP thin films in terms of mobile ion concentration (N_o_) and ionic mobility (µ) and demonstrate that N_o_ has a larger impact on device stability. We study the effect of small alkali metal A-site cation additives (e.g., Na^+^, K^+^, and Rb^+^) on ion migration. We show that the influence of moisture and cation additive on N_o_ is less significant than the choice of top electrode in PSCs. We also show that N_o_ in PSCs remains constant with an increase in temperature but μ increases with temperature because the activation energy is lower than that of ion formation. This work gives design principles regarding the importance of passivation and the effects of operational conditions on ion migration.

## 1. Introduction

MHPs with the general formula ABX_3_ (where A is a cation, B is a divalent metal ion, and X is a halide) are a family of semiconductor materials that have shown significant growth in efficiencies of PSCs from 3%, when they were introduced in 2009, to 25.7% in 2023, along with reduced manufacturing costs [1,2,3,4,5,6,7,8]. MHPs are known for their applications that span across various fields such as information display, electronic communication, health, and medical systems, and they are also widely used in solar cells, light-emitting diodes (LEDs), photodetectors, lasers, and X-ray scintillators because of their optoelectronic properties that include high absorption, high carrier mobility, and long diffusion lengths [9,10,11,12]. However, their commercialization is primarily hindered by instability in MHPs, and ion migration is one of the main factors that cause degradation in MHPs [13,14,15].

Ion migration is a challenge in MHPs because MHPs have ionic bonding as semiconductors that typically exhibit a soft crystal lattice and a mixed ionic-electronic behavior. This is different compared to conventional covalently bonded semiconductors [16,17,18] that are used for photovoltaic devices such as silicon, germanium, gallium arsenide, and cadmium telluride. Ion migration is a solid-state electrochemical phenomenon that happens in MHPs and is similar to the movement of ions in ionic conductors such as solid electrolytes for batteries and fuel cells. The mechanism is the redistribution of ionic defects, which happens both intrinsically and when subjected to external stresses such as light, heat, and an electric field. Vacancies ($V_{A}^{-}$, $V_{B}^{2-},V_{X}^{+}$) and interstitials ($X_{i}^{-}$) in the lattice form ionic defects or mobile ions that are predominantly halide in nature. Ions in MHPs form intrinsically due to low activation energies (E_A_), as shown in Table 1, and these ionic species are mobile in MHP and at the interfaces of MHP as they also have low migration energies [16,19,20,21].

It is reported that the concentrations of halide vacancies for different compositions in the MHP lattice vary between 10^15^ and 10^19^ cm^−3^, whereas concentrations of other ions such as MA^+^ have smaller concentrations (≤10^15^ cm^−3^). As the electric field is controlled by the ions with the highest concentrations, the rest of the ions need not be considered [22]. The presence and migration of these ionic species throughout the perovskite absorber, charge transport layers, and reactions with these layers and electrodes negatively affects the electrical properties and stability of MHPs [20,23].

In addition to electrostatic effects, ion migration in MHPs has an electrochemical effect that leads to MHP decomposition. It can cause reduction of uncoordinated *Pb*^+2^ ions to metallic lead and mobile *I*^−^ ions to react with atomic iodine (*I*) and methyl ammonium ions in MHPs to generate iodine (*I*_2_) both in darkness and under the influence of light, which not only can escape PSCs in the form of vapor, but also can trigger chemical chain reactions that accelerate perovskite degradation, as in Equations (1)–(7) [24]. MAPbI_3_ with inherent vacancies exist in the form of mobile ions, as in Equation (1):(1)$${CH}_{3}{NH}_{3}{PbI}_{3}⇌{PbI}_{2}+{CH}_{3}{NH}_{3}^{+}+I^{-}$$

These *I*^−^ ions can oxidize to form molecular iodine (*I*_2_) as in Equation (2)
(2)$$I^{-}+I^{-}⇌I_{2}+2e^{-}$$

Under exposure to light, *I*_2_ can undergo photolysis to form atomic iodine ($I^{•}$), and $I^{•}$ can react with $I^{-}$ to form a series of reactions, as in (3)–(5), to give $I_{2}$ as a by-product that again participates in cyclic reactions by (3)–(5).
(3)$$I_{2}\to 2I^{•}$$
(4)$$2I^{-}+2I^{•}\to 2I_{2}^{•-}$$
(5)$$2{CH}_{3}{NH}_{3}^{+}+2I_{2}^{•-}\to 2{CH}_{3}{NH}_{2}+2I_{2}+H_{2}$$

Under darkness, mobile *I*^−^ ions react with iodine to form tri-iodide ions (*I*_3_^−^), which react with methylammonium ions to form *I*_2_, as in Equations (6)–(7), and the generated I_2_ further participates in cyclic reactions.
(6)$$I^{-}+I_{2}⇌I_{3}^{-}$$
(7)$${CH}_{3}{NH}_{3}^{+}+I_{3}^{-}\to {CH}_{3}{NH}_{2}+I_{2}+HI$$

It has been shown that mobile ions in the MHP lattice of the perovskite can easily migrate to the charge transport layers. Ingress of halide ions into the electron transport layer (ETL) reportedly deteriorates the charge transport properties [25,26] and leads to the formation of accumulated charges that can interact with the migrated ions and form defects such as I^0^ and Pb^0^, leading to a gradual decay of the PSCs; halide ions can reduce the hole transport layer into a neutral state that reduces the conductivity of HTL and can also reduce the p-type doping of HTL to reduce hole extraction [27,28,29], leading to severe electronic losses in PSCs. Mobile halide ions also can migrate into and react with the top metal electrodes of PSCs, causing the irreversible degradation of the electrodes by forming metal halides; this loss of halide ions from the MHP lattice can lead to MHP decomposition in the form of volatile iodine-containing species (MAI, HI, *I*^−^) [30] that can further react with metal electrodes and corrode them, and the consequence is a reduction in the power conversion efficiency of the PSCs [20,31]. Under exposure to heat, MHPs have phase transitions with elevated temperatures above 85 °C due to the volatilization and loss of organic species from the MHP [32,33].

The most important parameters for quantifying ion migration in MHPs are mobile ion concentration (N_o_), which is defined as the number of mobile ions that are present in MHP; ionic conductivity (σ), which is defined as the capability of the ions to conduct electricity by the movement of ionic charge; and ionic mobility (μ), which is defined as a measure of how mobile the ions are under the influence of a voltage or electric field applied to PSCs. Ionic conductivity and ionic mobility can be related to each other based on their definitions, as in Equation (10), since σ depends on how effectively the ions can move (μ) in the MHP lattice. As shown in Table 2, commercialized solar cell technologies made from silicon are highly electronic in nature and have intrinsically fixed ions that are covalently bonded in the material for controllable doping to create a pn-junction. Hence, silicon does not exhibit any mobile ionic nature intrinsically as the ions are fixed in the crystal structure of silicon. The ions that we see in the silicon modules or silicon solar cells are the ones that are not intrinsic to the silicon; rather, they are being introduced extrinsically to the silicon modules based on encapsulation and also contamination due to human touch [34,35].On the other end of the spectrum, solid-state electrolyte technologies are designed for the purpose of ion conduction with minimal electronic conductivity, resulting in high N_o_ and very low μ. Note that this can be explained by the working mechanism of batteries, wherein their working depends more on the diffusion of ions rather than the drift of ions. Diffusion happens when there is a concentration gradient and is not dependent on μ, whereas drift of ions is directly dependent on μ. Since their working mechanism is diffusion, they do not have high μ. MHPs lie in between silicon solar cells and solid state electrolytes in terms of the number of ions present in them (N_o_) and the mobility of the ions and charges (ionic mobility and electronic mobility). The presence of such a high N_o_ and μ in PSCs is the reason why ion migration plagues the material system. These two parameters describe ion migration since more mobile ions facilitate ion migration and higher mobility leads to an increase in unwanted interactions in MHP lattices. Hence, for a PSC technology to be stable and reach the commercialization phase, there needs to be a reduction in N_o_ and μ without compromising the electronic nature.

In this work, we develop characterization techniques to decouple ion migration primarily into N_o_ and μ. We explain the methods used to quantify ion migration in MHPs by characterizing both MHP films and PSCs. We report on the changes in the ionic parameters as a function of MHP composition (including both cation and anions), top electrode, and operational conditions such as temperature, light, and moisture along with associated morphological changes.

## 2. Results

We developed a transient current measurement in the dark to measure N_o_ and used electrochemical impedance spectroscopy to measure σ. Then, μ was calculated using Equation (10). These measurements were performed on triple halide PSCs to compare with MAPbI_3_ PSCs, which are the most commonly used and are the simplest MHP composition. From Figure 1a, the N_o_ of the MAPbI_3_ PSCs [22] is in the range of 7 × $10^{16}$ to 5 × $10^{17}$ ${cm}^{-3}$, whereas the N_o_ of the triple halide PSCs is measured to be ~7 × $10^{15}$ ${cm}^{-3}$, a reduction of nearly 100× when compared to MAPbI_3_. This reduction can be attributed to the composition of MHP and the utilization of chlorine in the PSCs structure [36]. It has been shown that chlorine does not just remain at the interfaces but also gets incorporated into the MHP lattice by partially replacing iodine in the MHP lattice. The incorporation suppresses the iodine vacancy formation by ≥0.3 eV [37], an effect that we hypothesize leads to a reduction in N_o_. From Figure 1b, the μ of the triple-halide PSCs are about two orders of magnitude higher than in MAPbI_3_ based on the σ data from the EIS. Since the triple-halide PSCs are significantly more stable than MAPbI_3_, this result shows a clear correlation between reduced N_o_ and improved operational stability, based on the suppression of light-induced phase segregation and reduction in degradation [36].

We measured the N_o_, σ, and μ of triple-halide PSCs under temperature ramps between room temperature and 70 °C to understand the effect of temperature on the PSCs in terms of these parameters. From Figure 1c, the N_o_ of the PSCs does not have a noticeable change with increasing temperature. From Figure 1d, E_A_ is 0.14 eV, which is less than the E_A_ that is needed to form an iodine vacancy (Table 1), which implies that no new vacancies or defects are created, and this supports why no new mobile ions are generated. Note that this holds true up to 70 °C; higher temperatures were not studied in this work due to the possible diffusion of the top electrode through the C_60_ layer and into the MHP, which would lead to irreversible shunting and device degradation. However, from Figure 1d,e, it is evident that the σ and μ of the PSCs tend to increase with temperature. This shows that even if N_o_ remains constant, the ions are becoming more mobile with an increase in temperature. This is a phenomenon that is well-known, documented, and observed in other material systems such as solid-state electrolytes [38,39] and is also observed in MHPs [40]. With the ability to quantify mobile ions (N_o_, σ), and using the correlation between N_o_, σ, and μ in Equation (10), we are able to show for the first time that it is the mobility of the mobile ions that is changing with temperature. From these measurements, it can be concluded that this temperature range does not affect the intrinsic properties or defect density of MHPs. We hypothesize that since the ions are more mobile, they can lead to an increase in the number of interactions in MHP or faster migration to the interfaces of MHP and the transport layers, which can increase the degradation of the PSCs.

Passivation of MHP with potassium and rubidium has been shown to improve the optoelectronic properties and stability of PSCs because of the selective binding of K with bromine ions and Rb with iodine ions in the MHP lattice [41,42]. It was shown that the introduction of small cation additives in the MHP lattice can improve both efficiency and charge carrier mobility after passivation [41,42]. However, this improvement was dependent on the fraction of small cation additives that is introduced in the MHP lattice and how tolerant the MHP lattice is to the introduced additive. The electronic response of PSCs gets negatively affected once the fraction of additives crosses the tolerance level of MHP [41,42]. Based on the above observation, we hypothesized that the introduction of small cation additives could improve PSC stability due to a reduction in N_o_. Films with 5 mol% of the respective additives were fabricated with MAPbI_3_, a concentration or fraction that showed an increase in power conversion efficiency, superior photovoltaic performance, and moisture stability [41,42] for MHP with K and Rb as additives. As such, a 5 mol% addition of K, Na, and Rb was used. N_o_ measurements performed on these MHPs indicate the direct effect of small cation additives on the intrinsic properties of the perovskite absorber layer without any effects from other sources, such as interfaces with the transport layers and the composition of other PSC layers.

High resolution optical microscope images from Figure 2a–d indicate that the MHP morphology is affected by the additives. We hypothesize that this effect is due to a change in the crystallization where the MHP films with additives form a visible white intermediate phase during annealing briefly before crystallizing into a black perovskite phase, an effect that is not observed with control MHP films. We hypothesize that there is also an atomic-scale effect due to an increase in binding between the introduced additives and the mobile ions. K forms a bond with mobile halide ions primarily at the grain boundaries and interfaces of the MHP layer, an effect that could partially be explained by morphology change [41]. Rb forms bonds with mobile halide ions in the form of micron-sized crystals rich in Rb and iodine [41], changing the morphology when compared to the control sample. Na can form strong ionic bonds with mobile halide ions similar to the bonds we see in common salt (NaCl). The X-ray diffraction spectrum of the MHP thin films is shown in Figure 2f. The presence of MAPbI_3_ is confirmed as the dominant structural component of the MHP. In the case of the Rb additive MHP film, more peaks are observed, which are attributed to the higher quantity of residual PbI_2_ in the film. This effect could be related to the modified crystallization with the additive. The Na- and K-containing films, however, show little to no PbI_2_. N_o_ measurements of the MHPs show that these small cation additives reduce ion concentration in all cases by nearly a factor of 10. Figure 2e shows that the N_o_ of pure MAPbI_3_ thin film is ~2 × $10^{11}$ ${cm}^{-3}$, whereas the N_o_ of the films with additives such as KI, NaI, and RbI is ~6 × $10^{10}$ ${cm}^{-3}$, 3 × $10^{10}$ ${cm}^{-3}$, and 4 × $10^{10}$ ${cm}^{-3}$, respectively.

As the small cation additives proved to be effective in reducing the N_o_ in MHPs, complete PSCs were made with the same compositions. PSCs are more complicated systems because there are additional interfaces between the transport layers and the MHPs, along with the compositions of the transport layers to consider.

Observation of PSCs under a microscope, as shown in Figure 3a–d, depicted the same trend in morphology as that in MHP with an increase in binding in the MHP with the introduction of additives. The N_o_ measurements of these PSCs, as shown in Figure 3i, displayed a trend of increase in the concentration of PSCs with the presence of additives in the absorber layer.

PSCs that were exposed to moisture showed less uniform crystallization of the absorber layer and produced dendritic morphologies that had needle-like void structures, as shown in Figure 3e–h. These kinds of structures form when the crystallization process happens at a slower rate than required, which inhibits the nucleation process and also the grain-forming process [43]. The N_o_ measurements of moisture-exposed PSCs, as shown in Figure 3j, displayed a trend where the concentration in the PSCs increased with the introduction of the additives in the absorber layer. This trend or increase in concentration in PSCs is hypothesized to be because of the presence of needle-like structures in the morphology of these PSCs. These needle-like structures increase the defects at the grain boundaries and in turn increase the number of ions that are mobile in the lattice, increasing the N_o_ value.

Since even the PSCs with better morphology displayed a trend that is in exact contradiction to the trend observed in the MHPs, the same measurements were conducted on the PSCs with carbon as the top electrode. As shown in Figure 3k, N_o_ measurements with carbon electrodes exhibited a result that was similar to the trend observed with MHPs, i.e., a reduction in N_o_ with additives, except in the case of KI as an additive. We hypothesize that the increase in N_o_ with K as an additive is due to the selective binding of K, which tends to bind more with bromine than with iodine from the halide group and hence binds less when compared to Rb and Na, which increases N_o_ in PSCs.

## 3. Discussion

The comparison of the N_o_ of the triple-halide PSCs and MAPbI_3_ indicated that the reduction in N_o_ can improve operational stability. This correlation can be explained as follows: the reduction in N_o_ reduces the ion migration that is happening in the device when it is under operation.

Small cation additives such as KI, NaI, and RbI reduce the N_o_ value in the absorber layer, an effect that is evident from the N_o_ measurements in MHP films. It is also observed that after the introduction of additives, the MHP layer shows a change in morphology that can be attributed to slower nucleation, leading to the formation of fewer nuclei and hence less crystal growth [44,45,46] and potentially an atomic scale effect based on the selective binding of the small cation additives with the mobile halide ions. It is important to note that changes in N_o_ might not be dominated by morphology changes in the MHP layer since the same trend is observed in moisture-exposed PSCs (Figure 3i). We hypothesize that the trend that is observed with the N_o_ in PSCs is more due to changing the top electrode because N_o_ values measured from PSCs with C as the top electrode generally showed a similar trend compared to MHP films (Figure 3k).

For the same PSCs, if N_o_ measurements with the Ag electrode were in the range of $10^{14}$ ${cm}^{-3}$, the N_o_ measurements with the C electrode were in the range of $10^{11}$ ${cm}^{-3}$, a reduction of ~1000. This is much more significant than the ~10× reduction when incorporating small-cation additives into MHP films. We hypothesize that this difference in the N_o_ value is primarily caused by a passivation effect in the case of the C electrode when compared to the Ag electrode. The carbon ink used to form the C electrode consists of a small amount of acetone and ethanol, which can interact with MHP and facilitate the diffusion of C into the MHP. A passivation effect caused by C would reduce the number of defects and could increase binding to mobile halides, reducing the number of mobile ions in the area where the electrode is present. There is also a possibility that some of this can be attributed to the conductivity of C compared to Ag, an effect that needs to be more thoroughly studied. We believe that this is not as significant of an effect, however.

Future work is planned to better understand the mechanism for the change in N_o_ observed in the PSCs with additives and the effect of carbon electrodes. Carbon-based mesoscopic perovskite has previously shown an improvement in the stability of PSCs [47] and can be measured to understand the role of carbon in changes that were observed with N_o_. There are other methods for creating a carbon top electrode that are solvent-free and that would not interact or passivate the perovskite, an effect that can be compared with the measurements from solvent-based carbon top electrodes. We also intend to study more directly the effect of aging tests (specifically heat, light, and their combination) to see how N_o_, σ and μ change with time and exposure, similar to the clear variation in σ observed when the PSCs were measured in the dark vs. when they are measured under exposure to light (Figure 1f). Further characterization can also be used to correlate the degradation induced in PSCs due to both intrinsic changes in material chemistry and thermomechanical properties from external stresses such as light, heat, and moisture. Our intent is to elucidate the mechanisms and effects of the mobile ions on the stability of PSCs under operation and to effectively formulate strategies to mitigate these effects to improve PSCs’ commercial viability.

## 4. Materials and Methods

PSCs with a composition of pure methylammonium lead iodide (MAPbI_3_) along with some additives such as potassium iodide (KI), sodium iodide (NaI), and rubidium iodide (RbI) were fabricated to quantify and try to reduce ion migration. PSCs were made with Nickel Oxide (NiOx) as the hole transport layer (HTL) and C60 as the electron transport layer (ETL). The architecture of these solar cells is as follows: Glass/ITO/NiOx/MAPbI_3_ or MAPbI_3_ + additives/C60/Ag.

The substrate preparation steps are as follows: Glass with ITO (Indium tin oxide) coating is initially cleaned with an industrial-grade soap solution of extran and water at a ratio of 1:10 for 10 min in an ultrasonic cleaner. After sonication, the ITO glass is cleaned with deionized water and a brush to remove the residual soap, and it is then cleaned with isopropyl alcohol (IPA) and acetone for 10 min in that order, then cleaned with UV and ozone treatment for 15 min.

NiOx solution for depositing the HTL is prepared by mixing 1M NiNO_3_. (H_2_O)_6_ (99.999% pure) in 94% ethylene glycol (EG) and 6% of ethylene diamine (EDA); the vial is then placed in a vortex mixer, and the solution is mixed until it turns into a dark blue color indicating the solubility of the precursor into the solvent.

The perovskite precursor solution is made by mixing Methylammonium Iodide (MAI) (99% pure) and Lead Iodide (PbI_2_) (99.99% pure). A measure of 1 mL, 1:1 molar concentration solution is made by mixing 0.159 gm of MAI and 0.461 gm of PbI_2_ in a solvent of 4:1 Dimethyl Fluoride (DMF) and Dimethyl Sulfoxide (DMSO) with 800 µL DMF and 200 µL DMSO. A vortex mixer is used for mixing the precursor solution, which is then annealed at 80 °C until the precursor forms a clear solution.

Once the substrate preparation is done, PSCs are fabricated in a step-by-step process. The HTL is formed by depositing 50 µL of NiOx on the cleaned substrate and then spin coated at a speed of 5000 rpm at an acceleration of 2500 rpm/s for 30 s outside the glovebox and annealed at 315 °C for 1 h. The perovskite absorber layer is formed by depositing 200 µL of perovskite precursor on top of the samples and then spin coated at a speed of 2500 rpm with an acceleration of 1000 rpm/s for 30 s; 100 µL of Chloro-benzene (CHB) (99.8% pure) is dropped, which acts as anti-solvent at the half-time mark (15 s), and annealed at 100 °C for 30 min with a step at 50 °C until initial layer formation. The ETL is formed by evaporating 45 nm of C60 on the top of the samples inside the angstrom evaporator with the help of a custom mask, and the top electrode is made by evaporating 100 nm of silver (Ag) on top of the samples through a different mask.

MHP on ITO is fabricated following a similar recipe of perovskite absorber layer, as mentioned above. A measure of 200 µL of the precursor is then deposited on the ITO glass and spin coated for 30 s at a speed of 4000 rpm and an acceleration of 1000 rpm/s, and 100 µL of the anti-solvent is dropped on the sample at the half-time mark, and the samples are annealed at 100 °C for 30 min. The top electrode on the films is made using conductive carbon glue and is painted on the top of the MHP. To create a vertical stack kind of structure for the measurements, the MHP is scratched off at one edge of the sample to expose the ITO underneath it. Measurements are carried out by connecting probes to the top carbon electrode and the bottom ITO electrode.

Reduction of mobile ion concentration is achieved by adding small cation additives such as potassium iodide (KI), sodium iodide (NaI), and rubidium iodide (RbI) to the precursor inks in fixed concentrations, as they showed a reduction in hysteresis and improvement in efficiency in the PSCs [41,42]. The amount of the additive that needs to be added to the precursor ink for a particular concentration is determined by using Equation (8) listed below, where *x*—concentration of the additive to be added, [*k*]—amount of the additive to be added to precursor, and [*A*]—the amount of the precursor present [42].
(8)$$x=\frac{\left[k\right]}{\left[k\right]+\left[A\right]}$$

A measure of 1 mL 1 molar solution is made by mixing 0.166 gm of KI (99.995% pure), 0.149 gm of NaI (99.5% pure), and 0.212 gm of RbI (99.9% pure) with 4:1 DMF to DMSO solution, respectively. Then, 52.6 µL of the additive solution is added to the precursor solution to form a precursor with 5% additive, based on the calculations from Equation (1). MHPs with additives are formed by following the same spin-coating procedure that was mentioned above. The same precursors with 5% additives were used when fabricating PSCs with additives.

The ionic properties measured are ionic conductivity (σ), mobile ion concentration (N_o_), and ionic mobility (μ). All the measurements were performed with the help of all-in-one measurement equipment for photovoltaic devices and LEDs—PAIOS in the ambient. Ionic property measurements with the variation in temperature were performed with the temperature control stage and module (T96) from Linkam in integration with PAIOS. The T96 module from Linkam can increase the temperature of the stage in a controlled way with the help of resistive heating. The temperature measurements are performed from room temperature to 70 °C by ramping up the temperature of the stage by 5 °C at a time and then holding the temperature of the stage for 5 min so that the PAIOS can perform the measurements at that temperature before ramping up the temperature again.

Mobile ion concentration (N_o_) in PSCs and MHPs is extracted by using a transient current measurement on the samples in the dark [48]. As PSCs have both carrier separation and ion migration simultaneously, they have both drift and diffusion currents. The voltage applied to PSCs causes the mobile ions to drift into the contacts, leading to drift current (ionic current). The ionic charge (Q_ion_) of PSCs was measured by letting them equilibrate at 0.8 V in the dark (Figure 4a) based on previous work [22,49]. Then, the applied bias (V_app_) was removed, and the resulting dark transient current was measured until PSC reached an equilibrium steady state at 0 V. By doing so, we can probe only ionic currents since there is no electronic injection (diffusion current) at 0 V, and the electronic carriers initially present in the absorber at 0.8 V are swept away within the first few microseconds [49]. The drift (ionic) current is integrated over a time scale to obtain the Q_ion_ of the samples. After the Q_ion_ is obtained, by using an empirical relation [22] between charge, concentration, and applied voltages, we obtain the N_o_ of PSCs and MHPs as shown in Equation (9), where q—electronic charge, ε_o_—permittivity of free space, ε_r_—permittivity of the material, V_T_—thermal voltage, V_bi_—built-in potential, and Vapp—applied potential (0.8 V). Figure 4c shows the effect of V_app_ on the transient current. The maximum response of transient current was observed with V_app_ = 0.8 V, a result that is validated by previous work [22], and this was used for all N_o_ measurements in this work.
(9)$$Q_{ion}=\frac{\sqrt{qN_{o}\varepsilon_{o}\varepsilon_{r}V_{T}}}{8}\times \left[\sqrt{1+16*\left(\frac{V_{bi}}{V_{T}}\right)}-\sqrt{1+16*\frac{V_{bi}-V_{app}}{V_{T}}}\right]$$

The ionic conductivity (σ) of PSCs is measured by performing electrochemical impedance spectroscopy (EIS) [48]. In EIS, a small sinusoidal voltage of 0.05 V is applied to PSC, and the transient current is measured in the frequency range of 10 Hz to 10 MHz and analyzed according to a plot between real impedance vs. imaginary impedance (Nyquist plot). An equivalent circuit that can model the behavior of the device is fit onto the Nyquist plot to obtain the resistance and capacitance components that affect the ionic characteristics of the device [50,51,52]. The σ of the PSCs is then calculated by using the measured ionic resistance, MHP thickness, and the area of the electrode, as shown in Equation (10) [53], where *σ*—ionic conductivity, *t*—thickness of MHP, *R_i_*—ionic resistance, amd *A*—the area of the electrode.
(10)$$\sigma =\frac{t}{R_{i}\times A}$$

Ionic mobility (μ) is determined separately based on the relationship between ionic conductivity and mobile ion concentration determined by Equation (11), where *q*—electronic charge, *σ*—ionic conductivity, and *N_o_*—mobile ion concentration.
(11)$$µ=\frac{\sigma }{q\times N_{o}}$$

Triple-cation/triple-halide perovskites have a composition of (𝐹𝐴_0.75_Cs_0.25_)Pb(𝐼_0.8_𝐵𝑟_0.2_)_3_ + 2 to 5 mol% 𝑀𝐴𝑃𝑏𝐶𝑙_3_ [36] and a device structure of Glass/ITO/poly-TPD/PFN-Br/perovskite/C60/Ag. As the ionic species are mobile in the MHP lattice, it can lead to reactions with the photoexcited charges and can cause phase segregation to form individual halide-rich phases in mixed halide PSCs [54,55]. Triple-halide perovskites previously showed suppression of light-induced phase segregation, even at 100-sun illumination, and also showed a 2-fold increase in photocarrier lifetime and charge carrier mobility. They also showed less than 4% degradation in semitransparent top cells after 1000 h of maximum power point (MPP) operation at 60 °C [36].

## 5. Conclusions

In conclusion, this work seeks to fundamentally change the way in which ion migration in PSCs and MHPs is quantified. Most of the previous works quantified ion migration by subjecting PSCs or MHPs to stress tests such as exposure to heat, light, moisture and then correlating the degradation caused by the stress tests to the presence of ion migration by extracting imaging, compositional changes and activation energy changes after degradation. However, in this work we have developed a method to quantify ion migration in PSCs and MHPs in terms of N_o_ and µ. In this way, the ion migration can be quantified in situ and while under stress tests to make eventual degradation quantifiable. We showed that the introduction of small cation A site additives into MHPs reduced N_o_ and changes the morphology in a noticeable way when compared to control MHP films. However, the use of a carbon electrode for MHPs and PSCs played a substantially larger role in N_o_ even after introducing additives.

## Figures and Tables

**Figure 1 molecules-28-05026-f001:**
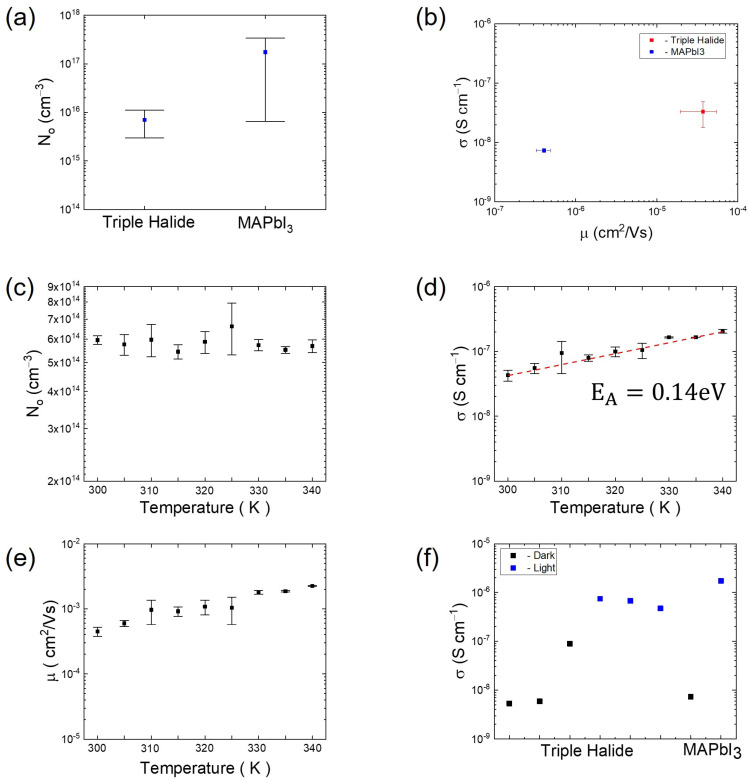
(**a**) N_o_ of triple halide PSCs in comparison with MAPbI_3_ PSCs, (**b**) σ and μ of triple-halide PSCs and MAPbI_3_ PSCs, (**c**) N_o_ of triple-halide PSCs with increasing temperature, (**d**) σ of triple-halide PSCs with increasing temperature, (**e**) μ of triple-halide PSCs with increasing temperature, (**f**) variation in σ with exposure to light vs. in the dark.

**Figure 2 molecules-28-05026-f002:**
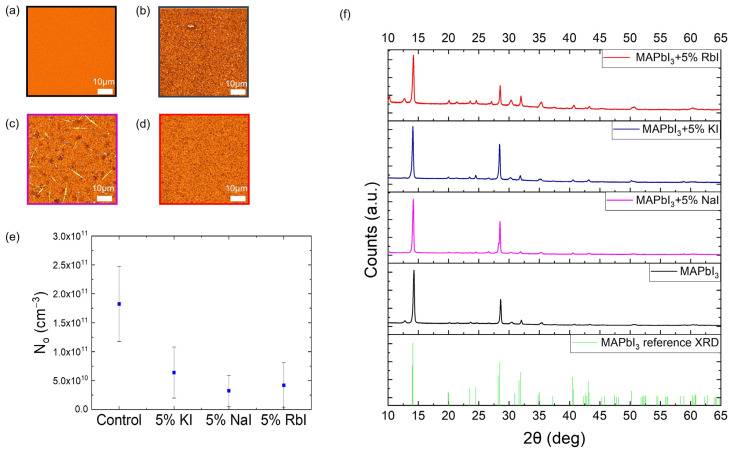
Optical microscope images of MHP thin films for control (MAPbI_3_) and with molar additives (5% KI, 5% NaI, 5% RbI) showing morphology changes in terms of needle-like structures and clustering in MHPs with additives. (**a**) Control, (**b**) 5% KI, (**c**) 5% NaI, (**d**) 5% Rb© (**e**) N_o_ of MHPs, (**f**) X-ray diffraction spectrum of the MHP thin films for control (MAPbI_3_) and with molar additives (5% KI, 5% NaI, 5% RbI).

**Figure 3 molecules-28-05026-f003:**
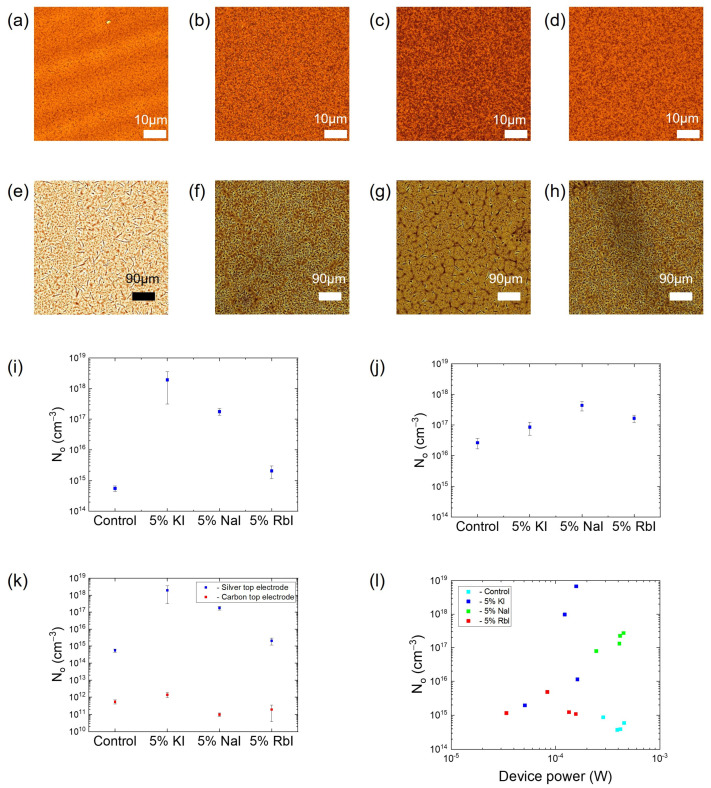
(**a**–**d**) Morphology of PSC with additives observed under microscope, (**e**–**h**) morphology of PSC exposed to moisture, observed under microscope. (**a**) Control, (**b**) 5% KI, (**c**) 5% NaI, (**d**) 5%©I, (**e**) control, (**f**) 5% KI, (**g**) 5% NaI, (**h**) 5% RbI, (**i**) no. of PSCs with additives (**j**), no. of PSCs exposed to moisture, (**k**) no. of PSCs with Ag vs. C as top electrode, (**l**) device power vs. no. of all PSCs.

**Figure 4 molecules-28-05026-f004:**
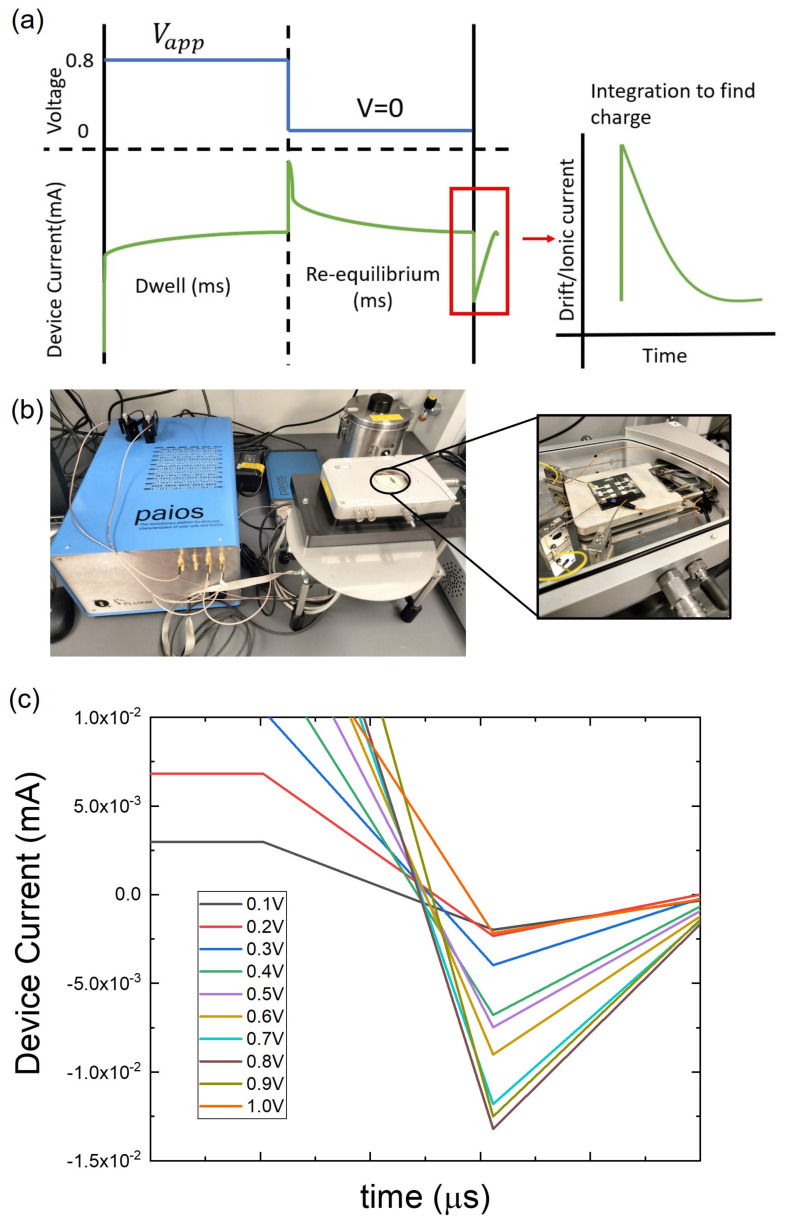
(**a**) Schematic of transient current measurements in the dark to obtain the ionic charge and N_o_; (**b**) setup of the equipment showing PAIOS and LINKAM stage (inset of the image) for the ionic measurements and temperature control, respectively; (**c**) variation in the transient drift current subject to the changes in the applied bias (V_app_) from 0.1 V to 1.0 V.

**Table 1 molecules-28-05026-t001:** Table showing activation energies of different vacancies in MHP lattice [16,19,20,21].

Vacancy	$\mathrm{Activation}\;\mathrm{Energy}\;(E_{A})$
$V_{I}^{+}$	0.58 eV
$V_{MA}^{-}$	0.84 eV
$V_{Pb}^{2-}$	2.31 eV

**Table 2 molecules-28-05026-t002:** Table showing N_o_, μ, and electron mobility of solar and battery technologies. Red indicates undesirable values and green indicates desirable values for solar cells.

Device	N_o_ (cm^−3^)	µ (cm^2^/Vs)	Electronic Mobility (cm^2^/Vs)
Solid State Electrolytes (Lithium Lanthanum Zirconium Oxide)	~5 × 10^18^ to 5 × 10^20^	~10^−10^ to 10^−14^	0.06
MAPbI_3_	2 × 10^17^	8 × 10^−6^	20 to 71
Triple Halide	5 × 10^15^	3 × 10^−4^	11 to 40
Silicon	0	0	~160

## Data Availability

Data available upon reasonable request.

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
