# Peer review of "Quantifying and Reducing Ion Migration in Metal Halide Perovskites through Control of Mobile Ions"

_molecules, 2023, doi:10.3390/molecules28135026_

Round 1

Reviewer 1 Report

In this study, authors reported that the influence of moisture and cation additive on mobile ion concentration (No) is less significant than the choice of the top electrode in PSCs. They also develop characterization techniques to decouple ion migration primarily into No and ionic mobility. This study gives design principles regarding the importance of passivation and the effects of operational conditions on ion migration. Authors should address following issues:

1- Figures 3 and 4 have a poor resolution to be published in a journal.

2- It is not clear why the authors chose 0.8 V to measure the ionic charge (Qion) of PSCs?

3- The given references are not updated and should be changed with recent studies. Some suggestions:

Instead of 3: 10.1039/D2EE01070D

Instead of 5: 10.1021/acsami.2c06110

Instead of 7: ACS Appl. Energy Mater. 2020, 3, 8, 7456–7463

Instead of 10: 10.1021/acs.jpcc.2c05882

4- Authors should revise this statement “Na can form strong ionic bonds with mobile halide ions because of strong electronegativity.”. As we know, electronegativity is the relative ability of an atom to gain electrons and become a negative ion. Sodium is clearly not very electronegative,.

5- There is a problem in view of some lines, page 4, line 123-124

6- In Figure 2, why No and ionic mobility values are 0 for Si?

7- There are some typos in the paper, please read and correct carefully.

8- Instead of Figure 1 and 2, authors should label them as Table 1 and 2, respectively.

There are some typos in the paper, please read and correct carefully.

Author Response

" Please see the attachment "

Reviewer 2 Report

The authors present a well-written paper about quantifying and reducing ion migration in perovskite. Halide perovskite is a very hot area nowadays and this paper will be interesting for many readers. However, some problems still need to be addressed before publication.

1. Figure 4 caption, please add the data obtained from microscope images.

2.In order to give the readers a bigger picture, more introduction about the perovskite materials should be added. The current introduction for perovskites is too simple. Their different applications such as photocatalysis, photodetectors and others should be introduced as well. The following related paper should be cited (https://pubs.acs.org/doi/abs/10.1021/acsmacrolett.0c00232; https://pubs.acs.org/doi/10.1021/acs.jpclett.7b01093).

3. Is that possible also to add some SEM images to show the morphology?

4. Figure 3, the picture resolution seems low. Please revise.

5. The XRD of the perovskites is missing. Please add XRD data for MHP and ion doped MHP.

Author Response

" Please see the attachment "

Round 2

Reviewer 1 Report

Authors addressed all the issues. I suggest the acceptance of this paper.

Reviewer 2 Report

The authors addressed all my questions very well